# Spectral clustering of risk score trajectories stratifies sepsis patients by clinical outcome and interventions received

Ran Liu[1,2]*, Joseph L Greenstein[1], James C Fackler[3,4], Melania M Bembea[3,4], Raimond L Winslow[1,2]*

[1]Institute for Computational Medicine, The Johns Hopkins University, Baltimore, United States; [2]Department of Biomedical Engineering, The Johns Hopkins University School of Medicine & Whiting School of Engineering, Baltimore, United States; [3]Department of Anesthesiology and Critical Care Medicine, The Johns Hopkins University School of Medicine, Baltimore, United States; [4]Department of Pediatrics, The Johns Hopkins University School of Medicine, Baltimore, United States

**Abstract** Sepsis is not a monolithic disease, but a loose collection of symptoms with diverse outcomes. Thus, stratification and subtyping of sepsis patients is of great importance. We examine the temporal evolution of patient state using our previously-published method for computing risk of transition from sepsis into septic shock. Risk trajectories diverge into four clusters following early prediction of septic shock, stratifying by outcome: the highest-risk and lowest-risk groups have a 76.5% and 10.4% prevalence of septic shock, and 43% and 18% mortality, respectively. These clusters differ also in treatments received and median time to shock onset. Analyses reveal the existence of a rapid (30–60 min) transition in risk at the time of threshold crossing. We hypothesize that this transition occurs as a result of the failure of compensatory biological systems to cope with infection, resulting in a bifurcation of low to high risk. Such a collapse, we believe, represents the true onset of septic shock. Thus, this rapid elevation in risk represents a potential new data-driven definition of septic shock.

*For correspondence:
rliu14@jhu.edu (RL);
rwinslow@jhu.edu (RLW)

**Competing interests:** The authors declare that no competing interests exist.

## Introduction

Sepsis and septic shock are the leading causes of in-hospital mortality (*Liu et al., 2014*), and are the costliest medical conditions in the United States (*Torio et al., 2016*). Improving outcomes in sepsis patients is therefore of great importance to public health. *Kumar et al., 2006* showed that every hour of delayed treatment in septic shock increased mortality by ~8%. More recent studies have corroborated this finding, though delayed treatment remains common in current practice (*Kumar et al., 2006*; *Martin-loeches et al., 2015*; *Ferrer et al., 2014*; *Levy et al., 2010*; *Kalil et al., 2017*). Consequently, a number of computational approaches for early prediction of sepsis and septic shock using electronic health record (EHR) data have been developed (*Henry et al., 2015*; *Nemati et al., 2018*; *Mao et al., 2018*; *Liu et al., 2019*). We recently developed a method for predicting patients with sepsis who are likely to transition to septic shock based on the hypothesis that there exists a detectable physiologically distinct state of sepsis, which we identify as the 'pre-shock state,' and that entry into this state presages the impending onset of septic shock (*Liu et al., 2019*). The pre-shock state was characterized by fitting a regression model to classify data from patients with sepsis who never develop septic shock from those who do. This model was used to calculate a time-evolving risk score

that can be updated each time a new measurement becomes available. When a patient's risk score exceeds a fixed threshold (time of early warning/prediction), we predict that the patient has entered the pre-shock state and is therefore highly likely to develop septic shock. Best performance achieved was 0.93 AUC, 88% sensitivity, 84% specificity, and 52% average positive predictive value, with a median early warning time (EWT) of 7 hr. We also showed that a patient's risk score at the first observation subsequent to threshold crossing was indicative of the confidence of a positive prediction. For some patients, this 'patient-specific positive predictive value' was as high as 91%.

The diverse patterns of sepsis make guideline-driven treatment difficult, as guidelines reflect the needs of the 'average' patient. Furthermore, treatments are not without risk. For example, high dosages of vasopressors have been associated with increased mortality (*Dünser et al., 2009*). Overuse of antibiotics is also a concern. *Minderhoud et al., 2017* showed that no evidence of bacterial disease is found in almost 30% of patients with suspected sepsis in the emergency department. A study by *Kelm et al., 2015* also showed that 67% of sepsis patients treated with early goal-directed therapy (EGDT) developed signs of fluid overload, which in turn increases risk of complications such as hypertension, pulmonary edema, and respiratory failure. It is likely that no single treatment policy is suitable for all sepsis patients, and thus, there has been interest in subtyping and stratifying sepsis patients in hopes of identifying phenotypes relating to patterns of treatment responsiveness. Individual biomarkers such as serum lactate have been used to stratify sepsis patients by mortality (*Mikkelsen et al., 2009*). Most recently, *Seymour et al., 2019* published a clustering study in which four types of sepsis patients were identified using the most abnormal value of 29 clinical variables observed in the 6 hr following hospital admission; these four types of patients differ in mortality and serum biomarkers of immune response.

In this study, we examine the temporal evolution of patient state as assessed using our previously-published method for computing patient risk of transition into septic shock. Previously we analyzed and interpreted this risk score from the perspective of threshold crossing (i.e. if, when, and/or how steeply a threshold crossing occurs), whereas here the time-evolution of risk in the hours following threshold crossing is hypothesized to yield further insight into patient state. We undertake a novel analysis of time-evolving risk scores to discover the distinct patterns of time-evolving risk that exist across patients. The application of spectral clustering to risk score trajectories reveals that patient risk trajectories diverge into four distinct clusters in the time window following early prediction of septic shock. Patients in these clusters stratify by outcome, as measured by prevalence of septic shock and by mortality. Moreover, these four clusters differ in the treatments received, as well as median time to septic shock onset (i.e. median EWT). In the highest-risk group, by time of early warning, fewer patients had been treated with vasopressors and been adequately fluid resuscitated than in the lowest-risk group. Time to septic shock onset was shorter in the highest-risk group as well. Using k-nearest neighbors to predict cluster membership, we are able to reliably assign patients to these risk categories based on their risk score trajectories.

Furthermore, we observe that the transition from sepsis to pre-shock on average occurs on a rapid time scale, with a sharp increase in risk occurring within 30–60 min immediately preceding time of early warning. This rapid change in the risk score is associated with rapid changes in values of systolic and diastolic blood pressure, lactate, and heart rate. Sepsis is defined as dysregulated immune response to infection (*Singer et al., 2016*). We hypothesize that the rapid transition of risk score results from and is therefore indicative of an abrupt failure of compensatory biological systems to cope with infection, resulting in a change in patient state trajectory that crosses a bifurcation from low to high risk. Such a collapse in compensation, we believe, represents the true onset of septic shock, and thus what we have previously referred to as the pre-shock state represents a potential new data-driven definition of septic shock. Previous studies of dynamical systems models of compensation of inflammatory responses have put forth a similar notion of disease progression in sepsis (*Reynolds et al., 2006*; *Chow et al., 2005*; *Cameron and Sleigh, 2019*; *Buchman, 1996*). The Infectious Disease Society of America's recently-published recommended revisions to the National Severe Sepsis and Septic Shock Early Management Bundle (*Klompas, 2020*) recognizes and acknowledges the ambiguity in determining the time of septic shock onset when applying current definitions, and states that there is a need for a clearer and reproducible definition of septic shock onset.

# Results

Using data from the 12 hr time window aligned at the time of early prediction, spectral clustering yielded four clusters. The clusters are labeled in order of descending septic shock prevalence (*Figure 1*; *Figure 1—figure supplement 1*; *Table 1*). Prior to early prediction, the distribution of risk score is homogenous in all four clusters. This is followed by an abrupt increase in risk score within the hour immediately preceding the time of early warning. Following this transition, risk scores diverge into four distinct clusters. Clusters 1 and 4 are of particular interest as they represent the highest and lowest risk patients, respectively, and thus we primarily analyze differences between these two groups of patients. *Table 1* shows that these clusters differ by septic shock prevalence (76.5% in the post-prediction high-risk cluster vs. 10.4% in the low-risk cluster) and mortality (43% in the high-risk cluster vs. 18% in the low-risk cluster). Higher risk clusters also have shorter times to septic shock onset (EWTs). In the high-risk cluster, the median elapsed time between the time of early prediction and time of septic shock onset is 9.8 hr, whereas in the low-risk cluster, it is 29.9 hr.

These four clusters also stratify based on the proportion of patients who have been treated by the time of entry into the pre-shock state (*Table 2*). Higher risk clusters have a lower proportion of patients who have been treated by the time of early prediction (i.e. higher risk clusters contain more patients with greater delays in treatment). At time of early warning, 7.8% of patients in the high-risk cluster are adequately fluid resuscitated, whereas in the low-risk cluster, 21.3% of patients are adequately fluid resuscitated. Similarly, in the high-risk cluster, 14.3% of patients have been treated with vasopressors by time of early warning, whereas in the low-risk cluster, 50.7% of patients have been treated.

The trajectories of physiological variables associated with patients in each of the risk score clusters evolve in a similar fashion to the risk score (*Figure 2*). Lactate, which has the greatest contribution to risk score (*Supplementary file 1B*), has a steep increase preceding time of early prediction

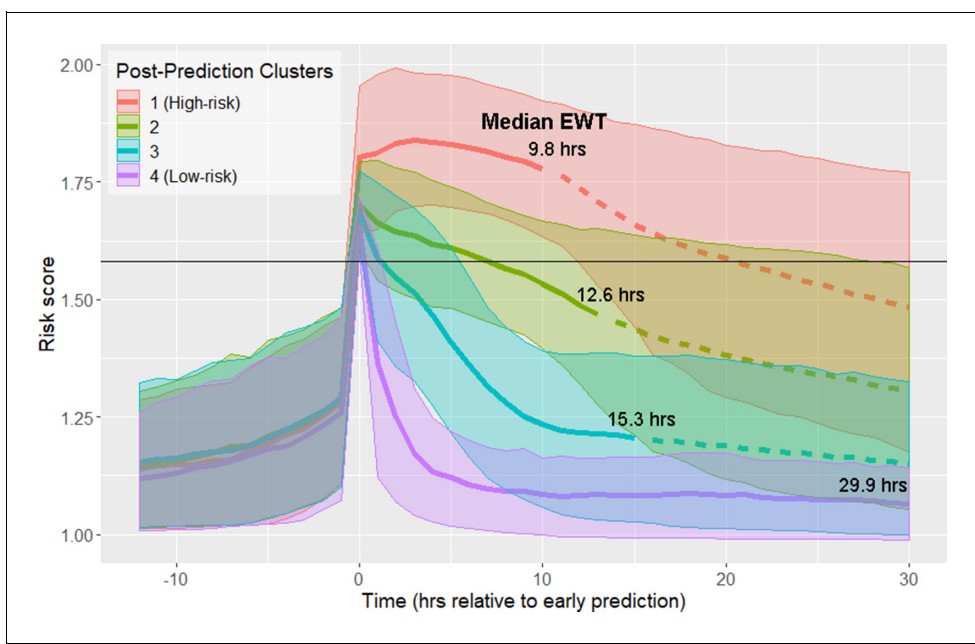

**Figure 1.** Risk score clusters obtained using spectral clustering on the 12 hr following time of early prediction. Time 0 represents $t_d$, time of early prediction. Bold solid and dashed lines indicate mean risk within each cluster. Each solid line becomes a dashed line at the cluster median EWT (indicated on figure). Shaded areas indicate one standard deviation from the mean. Black horizontal line indicates risk score threshold for early prediction.

The online version of this article includes the following figure supplement(s) for figure 1:

**Figure supplement 1.** Eigenvalues of Graph Laplacian of post-early prediction risk trajectories.

**Figure supplement 2.** Receiver operating characteristic curves for early prediction in eICU.

**Figure supplement 3.** Risk score clusters obtained using spectral clustering on the 12 hr following time of early prediction in the MIMIC-III database.

**Table 1.** Clusters in *Figure 1* stratify by septic shock prevalence, mortality, and time to septic shock onset (EWT). Clusters are numbered in descending order of septic shock prevalence.

| Post-Prediction cluster | Size | % Septic Shock | % Mortality | Median time to shock onset (EWT) |
|---|---|---|---|---|
| 1 (High-risk) | 1558 (17.2%) | 76.5% | 43% | 9.8 hr |
| 2 | 2672 (29.5%) | 46.3% | 34% | 12.6 hr |
| 3 | 3538 (39.0%) | 26.0% | 22% | 15.3 hr |
| 4 (Low-risk) | 1305 (14.4%) | 10.4% | 18% | 29.9 hr |

(*Figure 2A*). Systolic blood pressure has a steep decrease preceding time of early prediction (*Figure 2B*) and slight differences in mean heart rate (HR) are observed between clusters following time of early prediction (*Figure 2C*). The separation of these trajectories between the post-prediction low-risk and high-risk clusters, as quantified by Kullback-Leibler divergence, is not as great as that of risk score (*Figure 2—figure supplement 1*). Nevertheless, these rapid and coordinated changes of clinical feature trajectories mimic that seen in the risk score trajectory.

The results of *Table 2* demonstrate that the proportion of patients who have received interventions by time of early prediction varies between the low-risk and high-risk clusters. Early prediction immediately follows an abrupt change in physiological state, as reflected in a steep increase in risk score preceding threshold crossing. A greater proportion of low-risk patients than high-risk patients received intervention prior to this transition, indicating that cluster membership is influenced by intervention. Because, according to the Sepsis-3 criteria, septic shock is a treated state, with adequate fluid resuscitation and the administration of vasopressors as part of the labeling criteria, all septic shock patients receive intervention at some point; however, high-risk patients received one or more interventions at later times relative to the time of early warning as compared to low-risk patients. Moreover, median early warning time is shorter in the high-risk cluster than in the low-risk cluster.

There is not only a temporal difference in initiation of intervention between low-risk and high-risk patients, but also a difference in their physiological states prior to intervention. *Table 3* gives the mean values of 20 significantly different features (Wilcoxon rank-sum test, p<0.01, Bonferroni corrected) at 1 hr prior to the first instance of adequate fluid administration, and the eight significantly different features at 1 hr prior to the first instance of vasopressor administration. Not only do patients in the high-risk cluster receive treatment later than low-risk patients (on average), but patients in the high-risk cluster are also in a more severe state 1 hr prior to the first instance of each intervention than those in the low-risk cluster, with higher HR, lower blood pressure, higher lactate, and higher SOFA scores. The data demonstrate that the high-risk and low-risk clusters, the two most distinct in risk trajectory and outcome, differ in patient physiology immediately prior to the first instance of treatment with fluids or vasopressors.

One application of patient stratification via clustering is to predict whether or not any given patient is classified as high-risk. Patients predicted to be in the high-risk cluster are more severely ill than those predicted to be in the low-risk cluster. Classification of risk trajectories was performed using k-nearest neighbors, using between 0 and 12 hr of data following time of early prediction (*Figure 3*). As the amount of available post-threshold-crossing data increases, so too does the accuracy of classification of patient trajectories. Using a single observation subsequent to

**Table 2.** Proportion of patients in each of the four clusters who have received adequate fluid resuscitation or treatment with vasopressors by time of early prediction.

| Cluster | % Shock patients adequately fluid resuscitated | % Shock patients treated with vasopressors |
|---|---|---|
| 1 (High-risk) | 7.8% | 14.3% |
| 2 | 9.8% | 22.2% |
| 3 | 13.8% | 29.3% |
| 4 (Low-risk) | 21.3% | 50.7% |

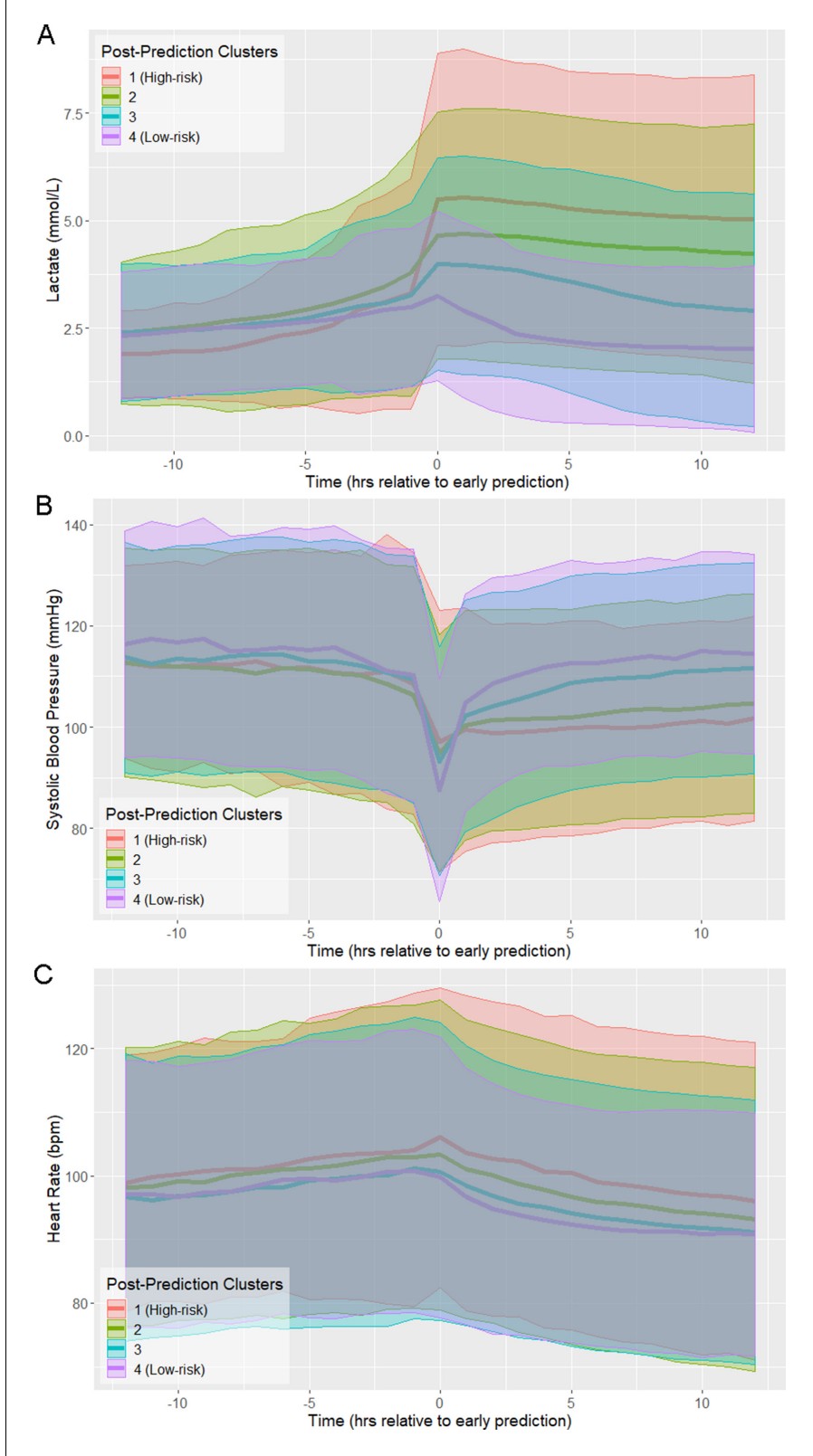

**Figure 2.** Physiological trajectories in (**A**) Lactate, (**B**) systolic blood pressure, and (**C**) heart rate for the 4 clusters of patients illustrated in *Figure 1*. Solid lines indicate the mean value of each feature within each cluster. Shaded areas indicate an interval of 1 standard deviation from the mean.

*Figure 2 continued on next page*

*Figure 2 continued*

The online version of this article includes the following figure supplement(s) for figure 2:

**Figure supplement 1.** Kullback-Leibler Divergence of Risk Score and physiological variables between the highest-risk and lowest-risk clusters in the window surrounding early prediction.

**Figure supplement 2.** Physiological trajectories in (A) Lactate, (B) systolic blood pressure, and (C) heart rate for the 3 clusters of patients illustrated in *Figure 1—figure supplement 3* in the MIMIC-III database.

**Figure supplement 3.** Kullback-Leibler Divergence of Risk Score and physiological variables between the highest-risk and lowest-risk clusters in the window surrounding time of early prediction in the MIMIC-III database.

time of early prediction, high-risk trajectories can be classified with 76% accuracy. With 10 hr of data (at the current rate of data collection), accuracy exceeds 95%.

All septic shock patients, as determined by the Sepsis-3 criteria, are treated at some point preceding or at the time of shock onset. Therefore, it is important to distinguish between physiological changes driven by disease progression (which can be observed prior to the first intervention), and changes in physiological state that may reflect response to intervention. We repeated our analysis of risk score trajectories by aligning risk scores about the time of first intervention. First intervention was defined as the earliest time antibiotics were ordered, vasopressors were administered, or adequate fluid resuscitation was achieved. Spectral clustering produced five clusters stratified by septic shock prevalence and mortality (*Figure 4*; *Figure 4—figure supplement 1*; *Table 4*). The highest risk cluster has a 75.9% prevalence of septic shock, and 42.2% mortality, whereas the lowest risk cluster has a 19.2% prevalence of septic shock, and 26.4% mortality. Furthermore, these clusters stratify by the mean time between entry into pre-shock and the time of first intervention (*Figure 4*). Differences in these times between clusters is statistically significant at the 99% confidence level (Wilcoxon rank-sum test, Bonferroni corrected). In the highest-risk clusters, there is the greatest delay between threshold crossing and the initiation of intervention. In the two lowest-risk clusters, initiation of intervention precedes entry into the pre-shock state on average. Note that risk score trajectories begin to diverge prior to the time of first intervention. For example, in the post-intervention high-risk (red line) cluster 1, the risk score is increasing more than 5 hr before the first intervention is

**Table 3.** Mean values of features which are significantly different at a 99% confidence level between the post-prediction high-risk and low-risk clusters at the time point 1 hr preceding (A) first instance of adequate fluid resuscitation or (B) first instance of vasopressor administration.

**A**

|  | High-risk | Low-risk |  | High-risk | Low-risk |
|---|---|---|---|---|---|
| HR (bpm) | 99.4 | 96.0 | BUN (mg/dL) | 39.8 | 33.2 |
| SBP (mmHg) | 107.2 | 112.4 | pH | 7.30 | 7.33 |
| DBP (mmHg) | 59.3 | 62.4 | $PaCO_2$ (mmHg) | 40.5 | 41.9 |
| MBP (mmHg) | 72.5 | 75.1 | Urine (mL/hr) | 5.6 | 3.2 |
| Resp (bpm) | 22.7 | 21.6 | Resp SOFA | 1.1 | 0.6 |
| $FiO_2$ | 66.2% | 61.2% | Nervous SOFA | 0.5 | 0.3 |
| GCS | 12.1 | 12.8 | Cardio SOFA | 0.4 | 0.1 |
| Platelets (k/μL) | 211.0 | 232.1 | Liver SOFA | 0.1 | 0.0 |
| Creatinine (mg/dL) | 2.2 | 1.9 | Coag SOFA | 0.5 | 0.3 |
| Lactate (mmol/L) | 4.6 | 2.9 | Kidney SOFA | 1.2 | 0.8 |

**B**

|  | High-risk | Low-risk |  | High-risk | Low-risk |
|---|---|---|---|---|---|
| HR (bpm) | 100.4 | 93.3 | pH | 7.28 | 7.30 |
| Resp (bpm) | 23.0 | 21.4 | $PaCO_2$ | 40.8 | 43.2 |
| CVP (mmHg) | 18.2 | 16.4 | Hemoglobin (g/dL) | 11.1 | 10.6 |
| $FiO_2$ | 71.2% | 64.2% | Hematocrit | 33.9% | 32.7% |

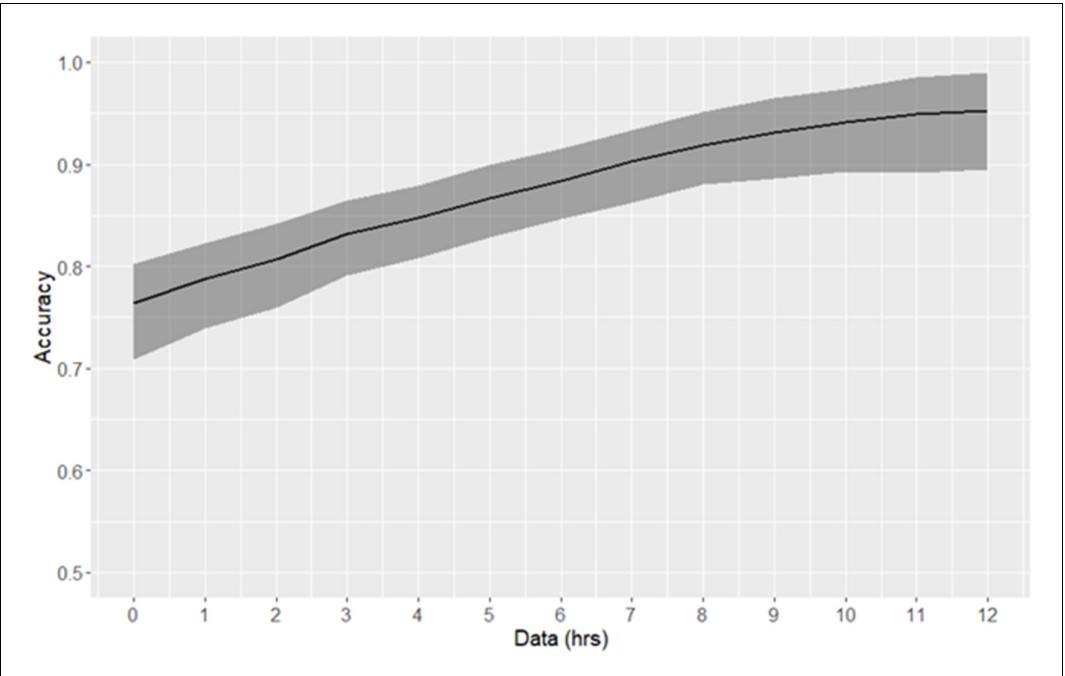

**Figure 3.** Risk trajectory classification accuracy. The duration of data used consequent to early prediction is specified by the x-axis. 90% confidence intervals, as empirically estimated using bootstrap, are indicated by the shaded area.

achieved. In the low-risk cluster 5 (magenta line), the risk score is decreasing approximately 3 hr before the first intervention is achieved. In general, the decline of risk scores follows the time of completion of first intervention. However, twelve hours following intervention, patient risk in most clusters is only slightly reduced compared to the value of risk score at the time of first intervention. These results demonstrate that, in the high risk clusters (clusters 1–3), the rate at which risk score growth slows and eventually declines following intervention is relatively slow when compared to rate of risk score growth prior to time of first intervention. Cluster membership, and hence patient risk, is therefore largely determined by patient state at time of first intervention. These results suggest that timely interventions are key to managing patient risk.

## Discussion

### Pre-shock state

In this study, we present an approach for stratification of sepsis patients that considers the evolution of their risk score over time, rather than their state at a fixed point in time. The observed divergence of risk score trajectories following entry into the pre-shock state indicates that there exists some

**Table 4.** Clusters in *Figure 4* stratify by septic shock prevalence and mortality.
A similar number of patients are in each cluster.

| Post-Intervention cluster | Size | % Septic Shock | % Mortality |
| --- | --- | --- | --- |
| 1 (High-risk) | 1506 | 75.9 | 42.2 |
| 2 | 1654 | 52.2 | 33.7 |
| 3 | 1758 | 38.5 | 26.9 |
| 4 | 2257 | 21.1 | 18.2 |
| 5 (Low-risk) | 1711 | 19.2 | 26.4 |

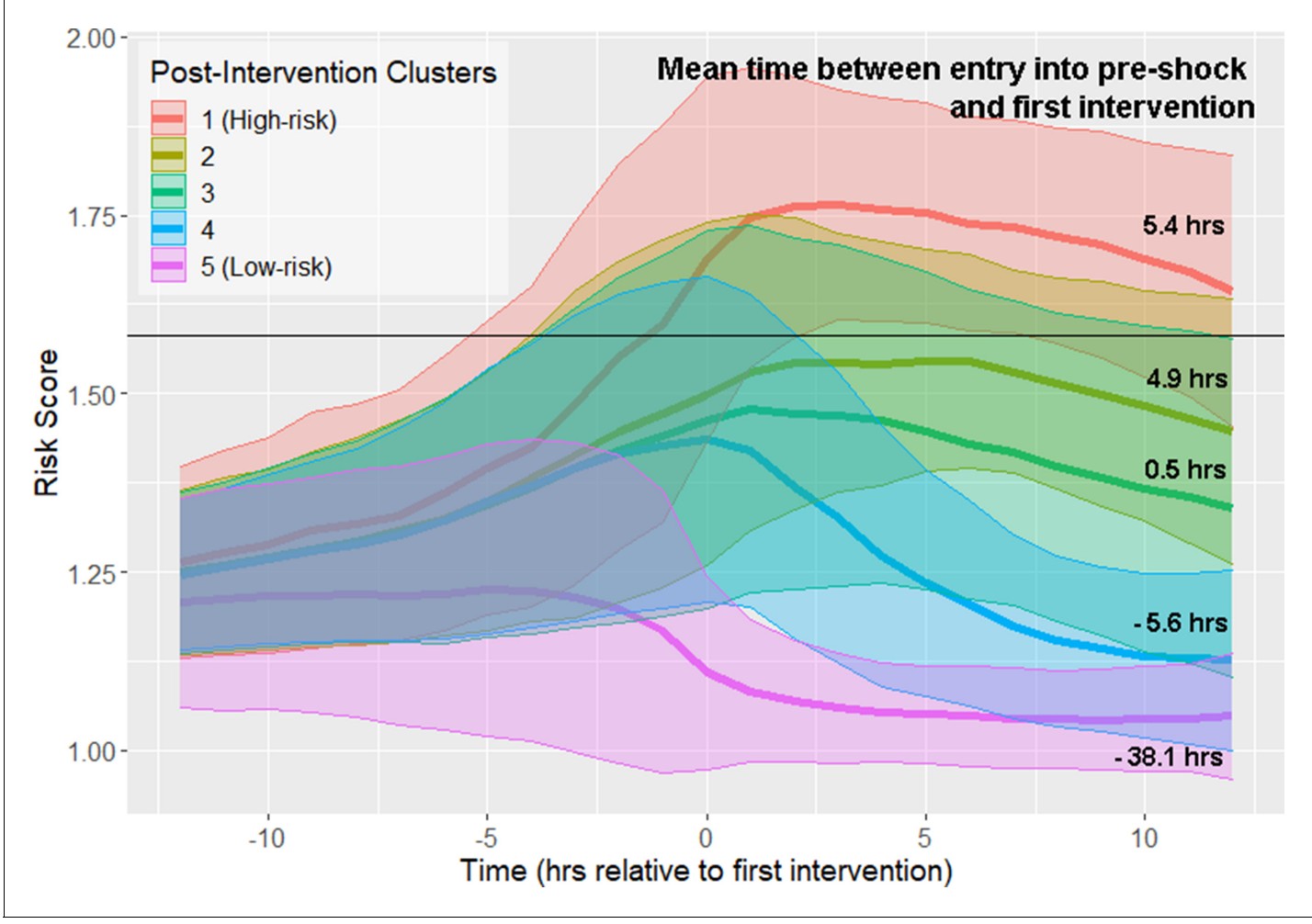

**Figure 4.** Risk score trajectories following the first instance of intervention. Threshold for early prediction is indicated by the horizontal line. Bold lines indicate mean risk within each cluster. Shaded areas indicate one standard deviation from the mean. The mean time within each cluster between entry into the pre-shock state and the time of first intervention is indicated on the right-hand side of the figure. A positive number indicates that the time of first intervention is after the time of threshold crossing, whereas a negative number indicates that the first intervention precedes entry into pre-shock. The online version of this article includes the following figure supplement(s) for figure 4:

**Figure supplement 1.** Eigenvalues of Graph Laplacian of post-intervention risk trajectories.

crucial time in sepsis patients when patient risk trajectory, and thus outcome, is determined well before these patients are clinically diagnosed as being in septic shock.

Previously, we hypothesized that there exists a physiologically distinct state of sepsis that precedes the imminent transition into septic shock (*Liu et al., 2019*). We referred to this state as the pre-shock state. *Figures 1* and *2* show that prior to entry into the pre-shock state, patients are indistinguishable on the basis of their risk scores or physiological variables. Risk trajectories diverge only upon entry into the pre-shock state (*Liu et al., 2019*). Entry into this state is rapid, with a large change in risk score occurring over a time window of 30–60 min. This rapid and statistically significant increase of risk occurs simultaneously with an increase of mean lactate and heart rate and a decrease of mean systolic blood pressure (*Figure 2*). These changes occur well before patients meet the Sepsis-3 definition of shock. We hypothesize that this rapid transition event reflects the collapse of an underlying biological control mechanism that, up to this time point, has helped limit the progression of sepsis. While this biological control mechanism has yet to be identified, we believe that its failure fundamentally defines septic shock. That is, when patients enter what we have previously called the pre-shock state, they are in fact in a state of septic shock. In the case of patients identified by clustering as high-risk, we believe the members of this cluster are in a state of shock on average

10 hr before they satisfy the current clinical definition of shock. Clearly, entry into the pre-shock state is sufficiently rapid that its detection will require intelligent automated monitoring of patients.

## Classification of trajectories

While this clustering study was performed retrospectively, one way in which these results may inform clinicians is by classifying new observations into one of the four clusters. Previously, we computed patient-specific positive predictive value by binning the first value of risk score subsequent to threshold-crossing into deciles, and computing the proportion of true positive patients in each bin (*Liu et al., 2019*). We showed that positive predictive value for patients with high risk scores relative to threshold could be as high as 90%. Here, we stratify risk trajectories using k-nearest neighbors. Since spectral clustering creates clusters that minimize within-cluster distances between data points, k-nearest neighbors using a similar distance metric is an appropriate choice of classifier, as it assigns to new points the most common label of its nearest neighbors. Using a single observation, we achieve 76% accuracy in classifying risk trajectories as high-risk or not. Intuitively, the certainty of this prediction increases with increasing length of observation; however, the rate at which performance increases is relatively slow. At 10 hr following time of early prediction, about half of observed patients would already meet the Sepsis-3 criteria for septic shock. This can be in part due to the infrequent rate of observation of important features. For example, the median time between observations for lactate, the most important feature in our risk model (*Supplementary file 1B*), is 11.2 hr (*Supplementary file 1E*). We therefore hypothesize that prediction performance would be improved, and reliable predictions could be made earlier if measurements of key features such as lactate were performed more frequently once risk score exceeds the threshold. *Vincent et al., 2016* have advocated for lactate measurements every 1–2 hr. The characteristics of the predicted cluster then serve to inform the clinician about patient prognosis, and overall severity of disease: patients whose risk trajectories are classified as high-risk are in grave danger of developing septic shock and/ or death, requiring the highest level of monitoring possible.

## Early interventions and intervention response

The data of *Figure 2* show that entry into the pre-shock state is a result of physiological change due to disease progression. Patient risk trajectories remain distinct following threshold crossing (*Figure 1*) and first intervention (*Figure 4*). *Kalil et al., 2017* suggest that responsiveness to treatment is determined by the baseline severity of sepsis . We corroborate this finding and show that physiological state prior to the initiation of treatment (*Figure 4*) is linked to risk trajectory after entry into the pre-shock state. The highest-risk patients had the greatest average delay between entry into pre-shock, and the initiation of treatment, whereas in the lowest-risk patients, initiation of treatment preceded entry into the pre-shock state (*Figure 4*). The physiology at time of first intervention, which in the majority of patients follows threshold-crossing (*Figure 1*), of patients in the low-risk cluster generally represents a less severe state than that of patients in the high-risk cluster. Patients in the low-risk cluster have lower heart rate, higher systolic blood pressure, higher GCS, and lower SOFA scores, all of which indicate a lower overall severity of disease (*Table 3*). Patients whose risk scores are high at the time of initiation of treatment have risk scores which remain high 12 hr later (and have correspondingly worse outcomes), and patients whose risk scores are low at time of first intervention have risk scores which remain low 12 hr later (*Figure 4*, *Table 4*). The data of *Figure 4* indicate that treatment is followed by an improvement in patients' condition, as assessed by risk score. However, this improvement is limited and has a slow time-course (*Figure 4*). The result is that patients in a more severe state of sepsis at time of first intervention generally cannot be rapidly rescued from a high-risk state, whereas initiation of treatment earlier in the disease progression generally mitigates risk over the remainder of the course of treatment. Therefore, it is perhaps critical that patients at risk of septic shock are treated in as early a stage as possible in order to achieve the most favorable outcomes. Achieving this goal of early treatment is not trivial. Some patients are admitted to the hospital or ICU already in a highly severe state of sepsis and treated immediately. The median time of entry into the pre-shock state for patients in the high-risk cluster precedes admission into the ICU by 135 min, and only follows admission into the hospital by 40 min. For patients in the low-risk cluster, this time is 48 min after ICU admission, and 8 hr after hospital admission. Many patients progress from low to high risk (as assessed by risk score) while heavily monitored in the ICU, however,

without the benefit of a risk scoring model, the combination of physiological changes that indicate such a progression may be difficult or impossible for clinicians to recognize (*Figure 2*), with the exception of lactate which is infrequently measured and often not ordered until there is already a clinical suspicion of sepsis (*Vincent et al., 2016*). The majority of transitions into the pre-shock state are not due to changes in lactate. A measurement of lactate coincides with threshold crossing (and thus can cause a change in risk score at that time) in 41.7% of patients. 56% of threshold crossings occur before any measurement of lactate is made. For these reasons, the availability to clinicians of a real-time risk score may be key in recognizing patients who would benefit from early stage interventions.

## Clustering algorithm

To cluster our time series data, we chose spectral clustering. Spectral clustering differs from algorithms such as k-means in that clustering is not based on distance from a centroid. Rather, clusters are chosen using a distance metric such that cluster members are close to one another. Whereas k-means results in linear boundaries between clusters, spectral clustering can produce clusters with nonlinear boundaries. Moreover, while we utilize Euclidean distance in this study, meaning that trajectories in the same cluster tend to have similar values at each time point, other distance functions can be used in spectral clustering. As our risk score trajectories are time series data, one possible alternative approach is to fit models to each time series, and define similarity between data points in terms of mutual information (*Jebara et al., 2007*).

## Generalizability of findings

Previously, we demonstrated the generalizability of our method for early prediction of septic shock, by training risk models on the MIMIC-3 database, and then applying them on the eICU database (*Liu et al., 2019*). We wish to be similarly confident that the results of this study generalize across patient populations. The eICU database used to build the classifier employed in this study consists of data from 208 hospitals in geographically disparate locations across the United States (*Pollard et al., 2018*). The consistency of our findings across this dataset makes it likely that the patterns discovered here apply across sepsis patients at large. To evaluate this, we repeated our methodology for clustering risk trajectories using the MIMIC-3 database (*Johnson et al., 2016*), from which we originally developed our methodology for early prediction of septic shock. We obtain the same principal findings of a rapid transition in risk at the time of threshold crossing, reflected in changes in physiology. Prior to this event, patient risk trajectories are homogenous, but diverge into clusters following this transition. These clusters stratify by prevalence of septic shock and mortality (*Figure 1—figure supplement 3*, *Figure 2—figure supplement 2*, *Figure 2—figure supplement 3*, *Supplementary file 1I*).

## Biological interpretation

Sepsis, and in particular, the competing interactions of inflammatory responses to sepsis and anti-inflammatory processes, have been conceptualized and modeled using dynamical systems (*Reynolds et al., 2006*; *Chow et al., 2005*; *Cameron and Sleigh, 2019*; *Buchman, 1996*). Various compensatory systems are responsible for maintaining homeostasis, represented by stable equilibria of the system, despite the perturbances of the sepsis disease state. However, when the disease progresses beyond a certain point, the properties of the system may change, and equilibrium points may lose their stability or disappear. In particular, *Reynolds et al., 2006* propose a model in which progression of sepsis results in a transcritical bifurcation, in which the stable equilibrium of compensated homeostasis (referred to as the 'health' state) loses its stability. One possible interpretation of our observations of patterns in patient risk trajectories is that processes such as those described by these dynamical systems models underly the transition from sepsis to septic shock. We find that prior to entry into the pre-shock state, patient risk trajectories change gradually, indicating that the patient is in a state of compensated homeostasis. However, an abrupt physiological event occurs which causes rapid change of risk. This phenomenon could be explained by the collapse of compensatory systems, resulting in the loss of homeostasis and stability. The shift of the underlying biological systems into an unstable regime would explain this rapid change in physiology.

## Spectral clustering

We choose the number of clusters (k) using the eigengap heuristic, choosing a value of k, such that the gap (difference) between the k-th and (k+1)th eigenvalues of the graph Laplacian is large relative to the gap sizes between other pairs of consecutive eigenvalues. The Davis-Kahan theorem (*Davis and Kahan, 1970*) guarantees that this choice of k results in a partition of data points which is robust to small perturbations. This means that the resulting clusters are stable even when small amounts of noise are introduced to the data, and thus, if the distribution of data that is being clustered is a good representation of the study population of interest, it is then likely that the same clusters will be found.

## Risk modeling

XGBoost (*Chen and Guestrin, 2016*) is a supervised learning method which uses gradient boosting of decision trees to build models for classification. Essentially, the model consists of the weighted sum of multiple decision trees; trees are added iteratively to the model, and subsequent trees attempt to minimize the error of previous models. Decision trees are able to learn nonlinear decision boundaries, and thus, XGBoost produces a nonlinear model of risk.

## Accounting for comorbidities

Comorbidities and pre-existing conditions potentially contribute to patient risk independently of their physiological state. 54% of sepsis patients in eICU have at least one comorbidity, as determined by a Charlson Comorbidity Index (CCI) greater than 0 (*Charlson et al., 1987*). Sepsis patients have a significantly greater CCI (Wilcoxon rank-sum test, p<0.01) than non-sepsis patients, as determined using their ICD-9 codes (*Quan et al., 2005*), though we did not find a significant difference in CCI between sepsis patients who did not develop septic shock, and sepsis patients who developed septic shock (*Supplementary file 1C*).

In the eICU database, pre-existing conditions are given in the *pasthistory* table. Of the 243 distinct comorbidities present, 26 are significantly associated with mortality, 15 are significantly associated with septic shock, and 3 (namely, COPD, hepatic encephalopathy, and cirrhosis/varices) have significantly different prevalence rates in the true/false positive cohorts for early prediction of impending septic shock (p<0.01, Fisher's exact test, Bonferroni corrected). The addition of the presence of these conditions as features in our risk model, however, did not significantly alter performance in early prediction. This is likely because the information contained within these features is not independent of the physiological data, and thus, any contribution of comorbidities to patient risk is captured by the model from physiological variables.

Prior studies have shown that between 34–66% of sepsis patients have at least one comorbidity, and that comorbidities had prognostic value in assessing risk of mortality in sepsis patients (*Artero et al., 2010*; *Oltean et al., 2012*), though *Innocenti et al., 2018* found that with the addition of physiological variables indicative of sepsis severity, such as lactate, comorbidities lost their independent prognostic value. Our findings are consistent with the results of the prior literature.

## Limitations

We algorithmically determine sepsis and septic shock labels according to the Sepsis-3 consensus definitions, and use the Surviving Sepsis Campaign (*Dellinger et al., 2013*) guidelines to determine when patients are adequately fluid resuscitated. Therefore, limitations inherent to these labels are also limitations of this study. One possibility is that the first instance of adequate fluid resuscitation may not give an accurate indication of treatment initiation for patients treated according to different protocols. This limitation may be mitigated in part by the proliferation of treatment protocols resembling EGDT for treatment of sepsis; if the majority of protocols sufficiently resemble EGDT in their administration of fluids, then the Sepsis-3 criteria may correctly identify patients who have been adequately fluid resuscitated, even under other protocols.

As previously mentioned, every observational study is limited by the potential for confounding factors. The task of inferring factors responsible in determining responsiveness to treatment is particularly difficult, as many variables besides the impact of intervention influence patient outcome. While it is possible to demonstrate association between variables, the only way to demonstrate causality is through randomized controlled trials; however, as *Liu et al., 2016* indicate, given the state of

equipoise in sepsis care, with many different treatment protocols heavily resembling one another, such a study is unlikely to be undertaken.

## Conclusion

We show that there exist four clusters of risk score trajectories in the time window immediately following entry into the pre-shock state. Prior to this time, patient physiological state is largely inseparable, but following entry into pre-shock, patient trajectory diverges into four strata, and these trajectories remain separate in the time window following pre-shock. One possible driving factor behind this divergence in patient trajectory is physiological state prior to initiation of treatment. The rapid change in risk score as patients enter pre-shock, and the relatively minute scale of change in physiological variables in this time window possibly indicate a general need for automated methods for early warning, as such changes may be imperceptible to human clinicians. Moreover, the abrupt nature of this transition possibly marks entry into septic shock. The transition from sepsis to septic shock, in general, is not a gradual event, but occurs on a rapid time scale, potentially as the result of loss of biological compensation mechanisms, leading to system instability. Further study into the biological mechanisms that underlie this sudden transition are important to understanding the nature of septic shock. Through use of risk score trajectories, it is now possible in retrospective studies to know at what critical point in time to look for biological signatures relating to this sudden transition of risk.

## Materials and methods

### Data extraction and processing

The eICU database (version 2.0) contains electronic health record (EHR) data collected between 2014 and 2015 from 200,859 ICU stays at 208 hospitals in the United States (*Pollard et al., 2018*). Data for these patients was extracted from the eICU PostgreSQL database using the RPostgreSQL package (*Conway et al., 2017*). The majority of entries in eICU are given as timestamp-value pairs where a label denotes the meaning of the value; these entries are spread across 31 tables. A total of 28 features (including heart rate, systolic blood pressure, lactate, urine output) are used in the calculation of our risk score trajectories. These entries are located in tables within the database, and an additional description column indicates the meaning of each entry. *Supplementary file 1A* specifies the location of each variable used in this study, along with associated description strings, and the column containing said description string. Multiple such strings may correspond to the same feature; for those features, all entries containing any of those strings were queried. For blood pressure measurements, invasive measurements took precedence over non-invasive measurements, i.e. if multiple entries existed for the same timestamp for a given patient, the invasive measurements would be kept and the non-invasive measurements at the same timestamp would be discarded. Otherwise, all such entries would be treated as equivalent. ICD-9 codes as specified by Quan et al. were used to compute the Charlson comorbidity index (*Charlson et al., 1987*; *Quan et al., 2005*).

### Infection criteria

Due to the lack of sufficient data for blood cultures in the eICU database, we determine suspected infection using the ICD-9 codes specified by *Angus et al., 2001*. Prevalence of the ICD-9 codes indicative of suspected infection most common in the eICU database are given in *Supplementary file 1G*.

### Labeling clinical states

The Third International Consensus Definitions for Sepsis and Septic Shock (Sepsis-3) were applied to patient EHR data to produce a time-series of clinical state labels (*Singer et al., 2016*). Sepsis patients are those with suspected infection and a Sequential Organ Failure Assessment (SOFA) score of 2 or higher (*Singer et al., 2016*; *Vincent et al., 1996*). Of the 200,859 ICU stays in eICU, 41,368 had suspected infection, as determined using the ICD-9 codes specified by *Angus et al., 2001*. Though *Seymour et al., 2016* recommend the use of concomitant orders for antibiotics and blood cultures, the limited availability of blood culture data in eICU necessitates the usage of ICD-9 codes. Septic shock patients are those with sepsis who have received adequate fluid resuscitation, require

vasopressors to maintain a mean arterial blood pressure of at least 65 mmHg, and have a serum lactate >2 mmol/L. Time of septic shock onset is determined as the earliest time when all conditions of septic shock are satisfied. Adequate fluid resuscitation was determined using the 2012 SSC guidelines (*Dellinger et al., 2013*), which is the version that was current at the time of collection of eICU data: adequate fluid resuscitation is defined as having received 30 mL/kg of fluids, or having attained fluid resuscitation targets of >0.5 mL/kg/hr urine output, mean arterial pressure of at least 65 mmHg, or a CVP of 8–12 mmHg. EGDT compliance was determined by computing whether or not adequate fluid resuscitation was achieved during at least one time point within the first 6 hr after ICU admission.

## Computing risk trajectory

Risk models were built according to the methods previously described, using 28 clinical features extracted from EHR data (*Liu et al., 2019*). We hypothesized the existence of a pre-shock state: that is, in patients who transition from sepsis to septic shock, there exists a physiologically distinct state of sepsis and entry into this state indicates that the patient is highly likely to develop septic shock at some future time. In order to characterize the pre-shock state, an XGBoost (*Chen and Guestrin, 2016*) regression model was trained using data from the sepsis clinical state in patients who do not go into septic shock, and data from a time window spanning 2 hr prior to septic shock onset to 1 hr prior to septic shock onset in patients who do transition to septic shock. Risk score trajectories were computed for each patient by applying this model at each time point of a window of interest using the most recently observed values of each feature.

Spectral clustering (see below) was applied to risk score trajectories aligned about a time point defined in two different ways. In the first (*Figures 1–2*), risk score trajectories and physiological time series were aligned at the time of entry into the pre-shock state, $t_d$, which corresponds to the time of occurrence of the first above-threshold risk score (*Liu et al., 2019*). The post-detection time window spanned from $t_d$ to $t_d$ +12 hr. In this window, risk score was computed at 1 hr intervals starting at $t_d$. In the second alignment method (*Figure 3*), risk trajectories were aligned about the time of the first intervention given to a patient, determined as the earliest of time of prescription for antibiotics, time of vasopressor administration, or the first time at which adequate fluid resuscitation was achieved.

## Clustering

Spectral clustering is a nonlinear clustering algorithm that generates clusters such that distances between points in the same cluster are minimized, and distances between points in different clusters are maximized (*Ni et al., 2002*). This is achieved by computing the eigenvectors of the graph Laplacian matrix. Each data point is a node on a weighted undirected graph: the diagonal entries of the matrix will be the number of connections each node has (usually a pre-specified constant), and the off-diagonal entries will be the connection weights. Connection weights, in this case, are given by applying a Gaussian kernel to Euclidean distance; if $w_{i,j}$ is an off-diagonal entry of the graph Laplacian, and $\delta(x_i, x_j)$ is the Euclidean distance between points $i$ and $j$, we have:

$$w_{i,j} = e^{-\delta(x_i, x_j)^2}$$

The eigenvectors of the graph Laplacian represent a nonlinear projection of the data into a new space, in which k-means or any other clustering method can be performed on the transformed data (*Figure 5*). By doing so, a solution to the semidefinite relaxation of the graph partition problem, which seeks to assign each node in a weighted undirected graph such that within-cluster weights are maximized, and between-cluster weights are minimized, is obtained (*Ni et al., 2002*). Therefore, risk score trajectories in the same cluster will have similar values of risk at each time point. In this study, the implementation of the *kernlab* R package was used (*Karatzoglou et al., 2004*).

Moreover, spectral clustering contains its own procedure for selecting k, the number of clusters, the eigengap heuristic. This procedure selects k such that the gap between the k-th and (k+1)-th eigenvalues of the graph Laplacian is large. Geometrically, by the Davis-Kahan theorem, this guarantees that the eigenvectors of the graph Laplacian are robust to small perturbations in the data (*Davis and Kahan, 1970*). Intuitively, this means that the results of spectral clustering for a selected

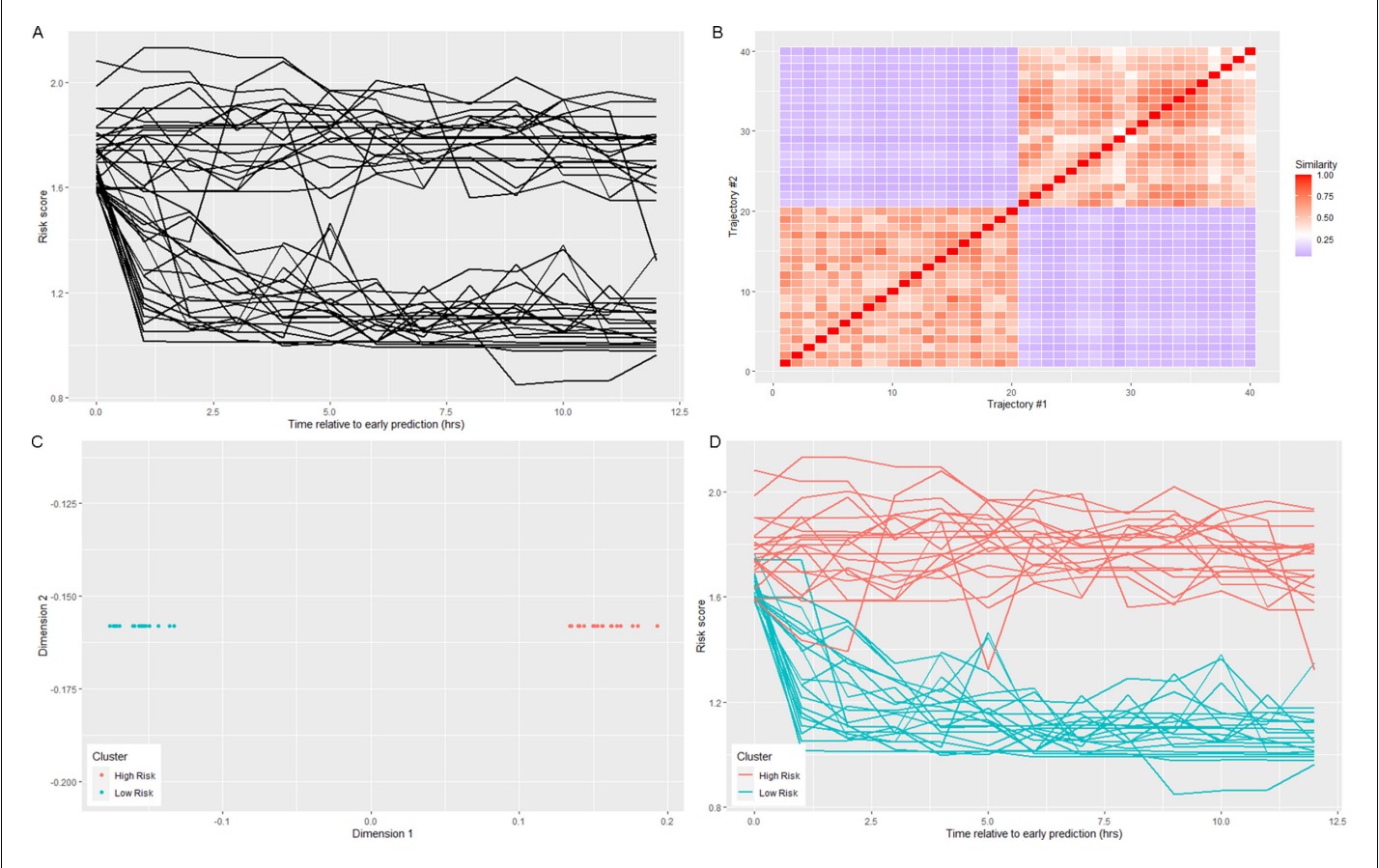

**Figure 5.** Visualization of spectral clustering of risk score trajectories, for a simple example with two clusters. Risk score trajectories (**A**) are clustered by using distances between trajectories (**B**) to project the data into a space in which they are easily separable (**C**). K-means clustering of the data in this new space yields clusters of risk scores (**D**).

value of k will be robust to small changes in the data, one common measure of goodness of fit for clustering algorithms (*Ni et al., 2002*).

## Classification of Trajectories

Classification performance for assigning new observations to clusters as a function of the number of post-detection samples (feature-vectors) was evaluated. First, clustering was applied to the entire dataset, obtaining cluster membership for every data point in the dataset; these clustering results are considered ground truth, and are used as labels for training and testing. 70% of the data was used as training data, and spectral clustering was performed on this training subset in order to generate training labels. K-nearest neighbors classification was used to predict cluster membership for the remaining 30% of the dataset, using the five nearest neighbors in the training set as measured by Euclidean distance. Performance was then evaluated against the ground truth clusters obtained on the entire dataset; clusters were numbered in order of descending prevalence of septic shock, and two clusters with the same order are considered equivalent.

## Acknowledgements

We would like to thank Drs. Nauder Faraday and Adam Sapirstein for valuable discussion. This work was supported by NSF EECS 1609038 and NIH UL1 TR001079.

## Additional information

### Funding

| Funder | Grant reference number | Author |
|---|---|---|
| National Science Foundation | EECS 1609038 | Raimond L Winslow |
| National Institutes of Health | UL1 TR001079 | Raimond L Winslow |

The funders had no role in study design, data collection and interpretation, or the decision to submit the work for publication.

### Author contributions

Ran Liu, Conceptualization, Software, Formal analysis, Investigation, Writing - original draft, Writing - review and editing; Joseph L Greenstein, Methodology, Writing - review and editing; James C Fackler, Conceptualization, Methodology, Writing - review and editing; Melania M Bembea, Conceptualization, Writing - review and editing; Raimond L Winslow, Conceptualization, Resources, Supervision, Funding acquisition, Methodology, Writing - review and editing

### Author ORCIDs

Ran Liu (iD) https://orcid.org/0000-0002-0866-9281
Melania M Bembea (iD) http://orcid.org/0000-0003-4984-520X
Raimond L Winslow (iD) https://orcid.org/0000-0003-1719-1651

### Decision letter and Author response

Decision letter https://doi.org/10.7554/eLife.58142.sa1
Author response https://doi.org/10.7554/eLife.58142.sa2

## Additional files

### Supplementary files

- Supplementary file 1. Supplementary tables.

- Transparent reporting form

### Data availability

All datasets utilized in this publication are publicly available; the eICU database can be accessed at https://eicu-crd.mit.edu/.

The following datasets were generated:

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
