## [Decision Letter]

**Acceptance summary:**

Intensive care medicine has recently been moving toward a patient centred, individualized or personalized care. The current manuscript falls nicely in this concept, by implementing a spectral clustering approach, which appreciates the fact that sepsis per se is not a definitive disease but a big, heterogeneous bucket, hence uniform structures, including scoring systems, cannot be applied with high certainty for the individual patient.

**Decision letter after peer review:**

Thank you for submitting your article "Spectral Clustering of Risk Score Trajectories Stratifies Sepsis Patients by Clinical Outcome and Interventions Received" for consideration by *eLife*. Your article has been reviewed by two peer reviewers, and the evaluation has been overseen by a Reviewing Editor and Eduardo Franco as the Senior Editor. The following individual involved in review of your submission has agreed to reveal their identity: Tamas Szakmany (Reviewer #1).

As is customary in *eLife*, the reviewers have discussed their critiques with one another. What follows below is a lightly edited compilation of the essential and ancillary points provided by reviewers in their critiques and in their interaction post-review. Please aim to submit within a couple of months a revised version that addresses these concerns directly. Although we expect that you will address these comments in your response letter we also need to see the corresponding revision in the text of the manuscript. Some of the reviewers' comments may seem to be simple queries or challenges that do not prompt revisions to the text. Please keep in mind, however, that readers may have the same perspective as the reviewers. Therefore, it is essential that you attempt to amend or expand the text to clarify the narrative accordingly.

Summary:

Risk stratification in the critically ill septic patients is of utmost importance as these patients may require the full arsenal of intensive care. Despite recent advances in intensive care medicine, mortality of septic shock is still high, around 40-60%, which also means that resources are spent in vain for those who died. High mortality is in part due to delayed recognition of disease severity, which can lead to late transfer to intensive care and delayed interventions. The present work advances our knowledge in this area by implementing a spectral clustering approach that takes into account the clinical complexity of the condition.

Essential revisions:

The Introduction and the Results section repeat a number of the learning points. It would be advisable to keep the Results only in this section, which could reduce the length of the manuscript.

The authors did not comment on the seemingly significant difference in the Charlson comorbidity index, and the presence of some of the chronic health conditions between the sepsis and septic shock group. As the patients have very similar physiology values in the "pre-shock" state, could these more chronic factors be partly responsible for sudden change in physiology? (i.e.: limited reserve due to chronic health issue manifests in more severe derangement?) We wonder if the authors would be able to incorporate this variable in the model, they would have an even better discriminator.

It is unclear for a clinician what exact value this risk-score would give at the bedside and whether it would need to be calibrated to different healthcare settings? Please elaborate on how one can generalize your findings.

The authors mentioned several times about the abrupt failure of compensatory biological systems. However, it is hard to understand what they meant. Could you provide a detailed discussion on the biological interpretation?

In the main text, the authors simply mentioned that they used their previously published method termed spectral clustering of risk score trajectories. To help readers understand their method, the authors may wish to add an illustrative figure to explain the logic of their method.

Please provide more detailed legends for each figure and table.

Figure 3 is about accuracy. Could you show the uncertainty in accuracy?

---

## [Author Response]

Essential revisions:The Introduction and the Results section repeat a number of the learning points. It would be advisable to keep the Results only in this section, which could reduce the length of the manuscript.

The Introduction has been revised so as to avoid repetition. Namely, the second to last paragraph of the Introduction has been trimmed so that it only presents an overview of the results, rather than going into the specifics of performance. This resulted in a reduction of length by ~100 words. However, the overall length of the manuscript has increased as a result of added Discussion sections, and the fact that supplementary text has been moved to the main body of the manuscript, in order to comply with formatting requirements.

The authors did not comment on the seemingly significant difference in the Charlson comorbidity index, and the presence of some of the chronic health conditions between the sepsis and septic shock group. As the patients have very similar physiology values in the "pre-shock" state, could these more chronic factors be partly responsible for sudden change in physiology? (i.e.: limited reserve due to chronic health issue manifests in more severe derangement?) We wonder if the authors would be able to incorporate this variable in the model, they would have an even better discriminator.

We are not entirely clear on the comment, and interpret it in the following way:

1) That there are differences in the distribution of the Charlson comorbidity index (CCI) between the sepsis (without septic shock) and septic shock patient cohorts that should be discussed;

2) That it is possible that chronic factors, i.e. comorbidities, influence the likelihood that a sepsis patient develops septic shock in a way that is not captured by the physiological variables used in the risk model;

3) That use of comorbidities as variables in the risk model could potentially improve the performance of early prediction of septic shock.

In response, we note that the first point is incorrect. Differences in the distributions of the Charlson comorbidity index between the sepsis and septic shock cohorts is not statistically significant, though there is a significant difference between the non-sepsis cohort and the sepsis cohorts. Recall that Supplementary file 1C states that the sepsis cohort has a mean CCI of 1.12, with a standard deviation of 1.5, and the septic shock cohort has a mean CCI of 1.14 with a standard deviation of 1.62.

However, the *frequency* of some individual comorbidities in Supplementary file 1D is different between the sepsis cohort and the septic shock cohort. Using these differences, we address the second point. In the course of developing this risk model, we did consider and study the addition of individual comorbidities as features prior to the submission of the manuscript. In the eICU database, there are records of patients’ pre-existing conditions in the *pasthistory* table. 243 distinct comorbidities exist in this table. Of these, 3 (namely, COPD, hepatic encephalopathy, and cirrhosis/varices) have significantly different prevalence rates in the true/positive cohorts for early prediction of impending septic shock (p<0.01, Fisher’s exact test, Bonferroni correction). However, adding the presence of these comorbidities as features in our risk model did not significantly change our early prediction performance. We believe that this is because the information contained in these features is not independent of the physiological data used as features in the current model.

With particular respect to Charlson comorbidities, these are computed using ICD-9 codes. While ICD-9 codes are useful for some aspects of retrospective analysis of patient cohorts, their use in a real-time algorithm or in a prospective setting are problematic because coding is not necessarily done in real-time. As their primary usage is for billing purposes, the time stamp of an ICD-9 code in current clinical practice could differ significantly from the actual time of the physiological events of interest and may therefore not be suitable for use in real-time algorithms.

Potential effects of underlying differences between the sepsis and septic shock cohorts are discussed in the subsection of the Discussion entitled “Early Interventions and Intervention Response”. An additional subsection has been added to discuss comorbidities, and the presence of chronic health conditions, in the Discussion subsection entitled “Accounting for Comorbidities”. Specifically, we discuss how in prior literature, comorbidities have been found to be highly prevalent in sepsis patients (as is the case in the eICU dataset), and have been shown to have prognostic value in sepsis patients (Artero et al., 2010, Oltean et al., 2012). However, prior literature has also shown that with the addition of physiological variables indicative of the severity of sepsis, comorbidities lose their prognostic value (Innocenti et al., 2018). Our findings are consistent with the results of these studies.

It is unclear for a clinician what exact value this risk-score would give at the bedside and whether it would need to be calibrated to different healthcare settings? Please elaborate on how one can generalize your findings.

The methodology for computing the risk score we utilize in this study is discussed in detail in our previous publication, “Data-driven discovery of a novel sepsis pre-shock state predicts impending septic shock in the ICU,” (Liu et al., 2019). The value of the risk score for early prediction of septic shock is that it gives clinicians early warning (many hours in advance) for patients with sepsis who are at high risk of impending onset of septic shock, and thus, a larger time-window for intervention that can potentially shorten treatment delays, and thus improve patient outcomes. However, early prediction is a binary label. The stratification of patients based on their risk score trajectories gives clinicians additional useful information regarding these predictions. Patients whose risk trajectories are in the highest-risk cohort are at the greatest risk of adverse outcomes, and knowing this provides physicians with useful information when they consider treatment approach. In practice, it may not be necessary to provide physicians with the numerical value of the risk score. Rather, a system may simply indicate high, medium, or low risk of septic shock. These points are discussed in the Discussion section, in the existing subsection, “Classification of Trajectories”.

In our previous paper, we discussed the generalizability of our method for early prediction, and showed that our method can be applied to datasets that are independent of those on which it was trained with only moderate changes in performance (peak performance of 0.85 AUC in eICU was achieved using models trained in MIMIC-3, compared to peak performance of 0.93 AUC testing on data from MIMIC-3). In practice, after a sufficient quantity of new data has been collected, the model can be retrained on data from the population in which it is being applied and the performance would be improved.

As for the generalizability of the findings presented in this paper in particular, the eICU database used in this study consists of data from 208 hospitals from across the United States. Because of the multiplicity of hospitals, the wide range of patients included in our study, and (none the less) the consistency of our findings across the dataset, it is likely that the patterns which we discovered in this study are generalizable to sepsis patients at large. Moreover, we are able to repeat our methodology of clustering risk trajectories in another dataset, the MIMIC-3 database in which we originally developed our methodology for early prediction of septic shock. We obtain the same principal findings of a rapid transition in risk at the time of threshold crossing, reflected in changes in physiology. Prior to this event, patient risk trajectories are homogenous, but diverge into clusters following this transition. These clusters stratify by prevalence of septic shock and mortality. These results are presented in Figure 1—figure supplement 3, Figure 2—figure supplement 2, Figure 2—figure supplement 3, and Supplementary file 1I. An additional subsection has been added to the Discussion entitled “Generalizability of Findings”. We have also been able to find these patterns in a separate study on pediatric patients (which is currently in the manuscript-preparation stage).

The authors mentioned several times about the abrupt failure of compensatory biological systems. However, it is hard to understand what they meant. Could you provide a detailed discussion on the biological interpretation?

Sepsis, and in particular, the competing interaction of inflammatory responses to sepsis and anti-inflammatory processes, have been conceptualized and modeled using dynamical systems (Reynolds et al., 2006, Chow et al., 2005, Cameron et al., 2003, Buchman et al., 1996). Various systems are responsible for maintaining homeostasis, and thus, stability of the system in the sepsis disease state. However, with further progression of the disease, the properties of the system change, and equilibrium points lose their stability. In particular, Reynolds et al. propose a model in which progression of sepsis results in a transcritical bifurcation, causing the stable equilibrium of health to lose its stability.

In the revised manuscript, we provide an interpretation of the findings of our study in the context of these conceptualizations of the sepsis disease state. Prior to entry into the pre-shock state, we find that patient risk trajectories change gradually, until an abrupt physiological event which causes rapid change in risk. These observations could be explained by the collapse of compensatory biological systems resulting in the loss of homeostasis. We’ve expanded the Discussion section accordingly: an additional subsection entitled “Biological Interpretation” has been added, discussing these points in greater detail as well discussing the prior literature. Mention of precedent for these ideas in the literature was also added to the Introduction, where we first introduce the concept of collapse of compensatory biological systems as a possible biological interpretation of our results.

In the interest of greater clarity, we also rewrote the subsection of the Discussion, “Early Interventions and Intervention Response”. Figure 4, its legend, and the corresponding subsection of Results have been revised with additional results, now including statistics on the mean time between first intervention and entry into the pre-shock state. These data help support the points we make in this revised Discussion regarding the impact of treatment delay on patient risk trajectory.

In the main text, the authors simply mentioned that they used their previously published method termed spectral clustering of risk score trajectories. To help readers understand their method, the authors may wish to add an illustrative figure to explain the logic of their method.

Calculation of the risk score is fully described in our previous paper, and is partially described in the Materials and methods section of this paper. We assume that the principal source of confusion is regarding the method of spectral clustering. We would like to clarify that the methodology presented in our paper is the application of spectral clustering for the stratification of risk trajectories in sepsis patients. The method of spectral clustering itself has existed for many decades.

For the sake of clarity in our particular application of the method, an illustrative figure has been added to the Materials and methods section (Figure 5). We also repeated the methodology of our previous study for early prediction of septic shock in the eICU dataset. These results are presented in Figure 1—figure supplement 2.

Please provide more detailed legends for each figure and table.

Legends have been expanded for figures and tables which were only briefly described. Namely, the legend for Table 1 has been expanded, the legend for Table 2 has been rewritten to be more descriptive, and the legends for Figures 2 and 3 have been expanded.

Figure 3 is about accuracy. Could you show the uncertainty in accuracy?

We used bootstrap to generate empirical confidence intervals for Figure 3, which has been updated in the Results section of the paper. The mean accuracy we obtain is slightly lower than originally reported; we obtain 76% accuracy using a single observation (as opposed to 80%), increasing to 95% after 10 hours.